# Segmentation of Optic Nerve and Lateral Ventricles in Low-Dose Non-Contrast CT with nnU-Net: A Pilot Study

**Hao Li**[1]                                                    HAO.LI.1@VANDERBILT.EDU

**Barite Gutama**[2]                                            BARITE.GUTAMA@VUMC.ORG

**Erin Abbott**[2]                                              ERIN.ABBOTT@VUMC.ORG

**Annika Coleman**[2]                                           ANNIKA.COLEMAN@VANDERBILT.EDU

**Matthew Pontell**[2]                                          MATTHEW.E.PONTELL@VUMC.ORG

**Ipek Oguz**[1]                                                IPEK.OGUZ@VANDERBILT.EDU

[1] *Computer Science, Vanderbilt University* [2] *Plastic Surgery, Vanderbilt University Medical Center*

**Editors:** Accepted for publication at MIDL 2025

## Abstract

In this study, we evaluate the nnU-Net, a self-configuring deep learning framework, for automated segmentation of the optic nerve and lateral ventricles from pediatric low-dose non-contrast brain CT scans. The segmentation performance is assessed using the Dice score via cross-validation. The overall mean Dice scores of 0.90 for the ventricles and 0.85 for the optic nerve indicate that nnU-Net can achieve promising segmentation performance in this context. In addition, we had human raters review the segmentations and provide revisions for even slight defects. The segmentations require few or no revisions, supporting the feasibility of structure-specific nnU-Net segmentation in low-dose CT and its potential to streamline annotation in future practical applications.

**Keywords:** Deep learning, intracranial pressure (ICP), non-invasive, craniosynostosis.

## 1. Introduction

Elevated intracranial pressure (ICP) is a common and serious complication in conditions such as craniosynostosis. The clinical gold standard for ICP estimation remains invasive intra-dural pressure monitoring, inserting a probe into the subdural space (Nag et al., 2019). This highlights the need for non-invasive approaches to identify at-risk patients early.

The optic nerve sheath diameter is a widely used imaging-based biomarker for estimating ICP via ultrasound (Robba et al., 2018; Abbinante et al., 2023), and ventricular volume from MRI has also been reported to have a potential association with ICP (Theodoropoulos et al., 2024; de Jong et al., 2012). However, volume measurement of these ICP-related structures has not been widely discussed in the context of low-dose non-contrast CT, which is routinely used in the evaluation of craniosynostosis for surgical planning (Pontell et al., 2024; Fearon, 2014; Chaij et al., 2025).

Human annotation of 3D CT scans is both time-consuming and resource-intensive. Deep learning approaches may offer a reliable way to accelerate this process by providing initial predictions, reducing the need to annotate from scratch. While deep learning approaches have been widely applied to 3D CT data (Isensee et al., 2021; Wasserthal et al., 2023; Li et al., 2024; Liu et al., 2024; Podobnik et al., 2024), the segmentation of specific structures such as the optic nerve and ventricles remains relatively underexplored, especially in low-dose non-contrast CT scans, where limited soft tissue contrast poses additional challenges.

In this paper, we evaluate the performance of deep learning-based segmentation of the optic nerve and ventricles in low-dose CT scans, using nnU-Net as a representative baseline. The segmentations are reliable, requiring only minor adjustments during human review. These results suggest that deep learning methods can effectively support more efficient annotation workflows by providing initial segmentations.

## 2. Materials and Methods

**Datasets and preprocessing.** Two labeled CT datasets for optic nerve (n=31) and ventricles (n=22) are used. The resolutions range from $0.23 \times 0.23 \times 0.5$ to $0.53 \times 0.53 \times 0.75 \text{mm}^3$ (optic nerve) and $0.35 \times 0.35 \times 0.3$ to $0.52 \times 0.52 \times 5 \text{mm}^3$ (ventricle). The preprocessing includes isotropic resampling and z-score normalization based on foreground voxels.

**nnU-Net.** We adopt the generic 3D segmentation configuration from nnU-Net (Isensee et al., 2021), a widely used framework for medical image segmentation. It utilizes a six-level 3D U-Net architecture (Çiçek et al., 2016) with deep supervision (Zeng et al., 2017), which introduces auxiliary losses at intermediate layers to improve training stability and encourage consistent feature learning across scales. The model is trained from scratch using only the provided annotations, without any pretraining or external data.

**Evaluation.** Each dataset is evaluated using four-fold cross-validation, with Dice scores reported for evaluation. Additionally, to provide a complementary assessment of segmentation quality, the outputs generated by nnU-Net were reviewed by human annotators. They evaluated the segmentations and revised as needed for even minor defects. The initial ground truth labeling and refinement were conducted by the same individuals but two months apart, and files were anonymized and randomized for revisions to avoid bias.

## 3. Results and Discussion

**Results.** Tab. 1 presents the Dice scores for nnU-Net segmentation using four-fold cross-validation. For each structure, the variation in mean Dice scores across folds is modest, and the standard deviations within each fold remain consistently low. These results suggest that the deep learning framework can effectively segment these anatomical structures from low-dose non-contrast CT scans, despite the limited soft tissue contrast. The revision results are shown in Fig. 1. While some folds contain a relatively large number of revised cases, most involve only minor Dice changes, suggesting most original segmentations are satisfactory.

**Discussion.** Manually annotating CT scans is inherently challenging due to limited soft tissue contrast, and minor defects can bias downstream analyses for ICP, especially for optic nerve. In Fig. 1(c), the result was slightly under-segmented (green arrow) compared

Table 1: Dice scores (*mean ± std. dev%*) for nnU-Net predictions with cross-validation.

| Structure | Fold 1 | Fold 2 | Fold 3 | Fold 4 | Overall | Revision[†] |
|---|---|---|---|---|---|---|
| Optic nerve | 83.87±0.05 | 86.14±0.07 | 83.63±0.05 | 86.27±0.05 | 84.94±0.06 | 84.69±0.05 |
| Ventricle | 90.43±0.03 | 86.41±0.08 | 92.16±0.04 | 92.91±0.02 | 90.29±0.05 | 90.27±0.05 |

[†] Dice between human-revised nnU-Net segmentations and the ground truth.

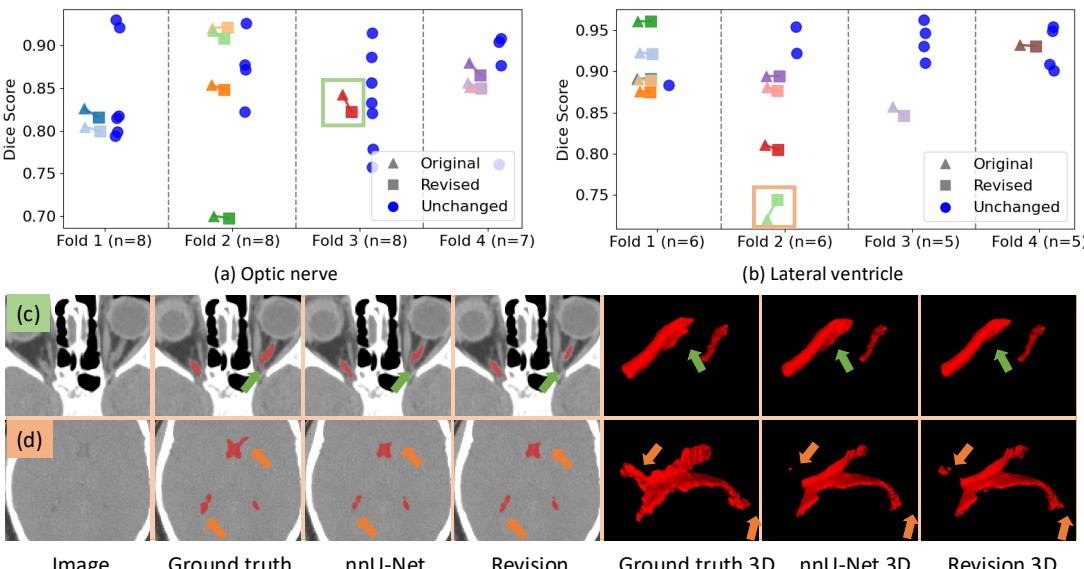

Figure 1: (a) and (b) show the Dice scores before and after revisions for the optic nerve and ventricle, respectively. Original-revised pairs are color-coded, and the number of test subjects per fold is indicated. (c-d) highlight the subjects with the largest disagreement between the original results and the revisions. (c) shows the ground truth annotation was likely over-segmented. (d) shows topological defects.

to the 'ground truth' annotation. However, during revision, the segmentation was even further trimmed in this area, suggesting the initial ground truth annotation may have been oversegmented. This indicates that the model may help highlight ambiguous boundaries that are difficult to assess visually and may have been mislabeled in the initial annotation. Fig. 1(d) shows that the model produces a disconnected segmentation in the anterior region of the lateral ventricle, which remains unchanged during revision. This discrepancy likely arises from an over-reliance on model output and the challenges of annotating CT images in 2D slices, where ambiguous details and 3D connectivity are easily overlooked. These findings suggest that an inter-rater review process may be necessary to reduce such discrepancies.

## 4. Conclusion

In this paper, we evaluate nnU-Net for optic nerve and ventricle segmentation from low-dose non-contrast CT scans, where soft tissue contrast is limited. The results show consistently reliable segmentation performance across cross-validation folds, as further evidenced by minimal revisions by human raters. The approach may help accelerate the annotation process instead of starting from scratch. Future work will be expediting large-scale segmentation to create growth curves against which case populations can be compared, in order to non-invasively estimate potentially pathologic changes in the intra-cranial milieu.

**Acknowledgments.** This work was supported by Vanderbilt Institute for Surgery and Engineering (VISE) Seed Grant.

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
