# OpenReview forum: "Segmentation of Optic Nerve and Lateral Ventricles in Low-Dose Non-Contrast CT with nnU-Net: A Pilot Study"
_MIDL.io/2025/Short_Papers — MIDL 2025 - Short Papers_

### Official Review · Reviewer_1dCC · 2025-04-19

**Rating:** 5
**Confidence:** 4

**Summary:**

The paper proposes to use standard nnUNet for two important structures of interest in low-dose non-contrast CT images. Results on a private dataset reveal that nnUnet can give high performance with minimal correction by an expert.

**Strengths:**

* Good introduction and motivation.
* Results are well presented.
* Experimental study with doctors is really insightful.
* Good discussion section which clearly describes the limitation of the current study.

**Weaknesses:**

My only point of concern is the employed evaluation metric. Although it is a standard metric for segmentation, it has its own limitations [1]. Employing other metrics like HD or Surface Dice might be a good idea.

[1] Maier-Hein, L., Reinke, A., Godau, P., Tizabi, M.D., Buettner, F., Christodoulou, E., Glocker, B., Isensee, F., Kleesiek, J., Kozubek, M. and Reyes, M., 2024. Metrics reloaded: recommendations for image analysis validation. Nature methods, 21(2), pp.195-212.

---

### Decision · Program_Chairs · 2025-05-01

Accept